# The double protonation of dihapto-coordinated benzene complexes enables dearomatization using aromatic nucleophiles

Justin T. Weatherford-Pratt [1], Jacob A. Smith[1], Jeremy M. Bloch [1], Megan N. Ericson[1], Jeffery T. Myers[1], Karl S. Westendorff [1], Diane A. Dickie [1] & W. Dean Harman [1] ✉

Friedel-Crafts Arylation (the Scholl reaction) is the coupling of two aromatic rings with the aid of a strong Lewis or Brønsted acid. This historically significant C–C bond forming reaction normally leads to aromatic products, often as oligomeric mixtures, dictated by the large stabilization gained upon their rearomatization. The coordination of benzene by a tungsten complex disrupts the natural course of this reaction sequence, allowing for Friedel-Crafts Arylation without rearomatization or oligomerization. Subsequent addition of a nucleophile to the coupled intermediate leads to functionalized cyclohexenes. In this work, we show that by coordinating benzene to tungsten through two carbons (dihapto-coordinate), a rarely observed double protonation of the bound benzene is enabled, allowing its subsequent coupling to a second arene without the need of a precious metal or Lewis acid catalyst.

Whereas organic alkenes are widely known to undergo a reaction sequence of protonation followed by nucleophilic addition (i.e., alkene electrophilic addition), such a process is generally not accessible to benzenes, owing to the highly stabilized aromatic ring. However, we recently demonstrated that this reaction sequence was not only possible for a dihapto-coordinated ($\eta^2$) benzene complex (Fig. 1)[1,2], but that the resulting $\eta^2$-diene complex (**3**) could participate in a second protonation/nucleophilic addition sequence to form the corresponding cis-3,6-disubstituted cyclohexene complex (**4**)[2]. The range of nucleophiles that can be added in this manner includes cyanide, enolates, Grignard reagents, amines, and alkoxides[2]. However, the highly π-basic nature of the tungsten system required to coordinate and activate the benzene ring also stabilizes the $\eta^2$-arenium intermediate (**2**), thereby diminishing its ability to react with so-called "π-nucleophiles" such as arenes and alkenes[3]. We posited that a different strategy, in which the benzene was first double-protonated, might be possible. The resulting dicationic species (**5**), if accessible, would be highly electrophilic, and could be capable of reacting with much

milder nucleophiles than its monoprotonated precursor **2**. The resulting π-allyl species (e.g., **6**–**8**) would then be positioned to react with a second nucleophile to generate a 3,6-disubstitued cyclohexene complex (e.g., **9**–**11**). Herein we show a new approach to dearomatization of benzenes in which coordination to a tungsten complex enables the double protonation of the bound arene and subsequent coupling to a second aromatic molecule. The resulting allyl species can then be combined with a "hard" nucleophile resulting in novel cis-3,6-disubstituted cyclohexenes.

## Results

### Double protonation of benzene

We initially focused on the parent benzene complex, WTp(NO)(PMe$_3$) ($\eta^2$-benzene) (**1**; Tp = (trispyrazolyl)borate), which we have previously shown could be protonated by diphenylammonium triflate (pK$_a$ -1; DPhAT) to generate the $\eta^2$-arenium complex **2** (Fig. 1)[1]. When the $\eta^2$-arenium species **2** is treated with a CD$_2$Cl$_2$ solution of triflic acid (HOTf; T = 0 °C), the [1]H NMR spectrum reveals that a second protonation

[1]Department of Chemistry, University of Virginia, Charlottesville, VA, USA. ✉e-mail: wdh5z@virginia.edu

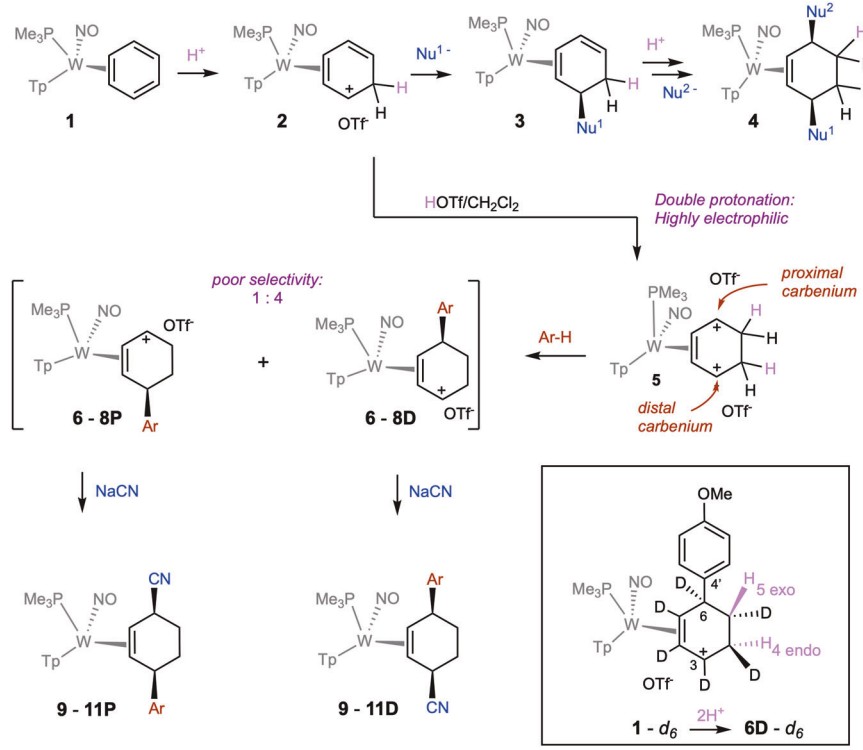

**Fig. 1 | The tungsten-promoted double-protonation of benzene followed by electrophilic aromatic substitution (EAS), and addition of cyanide ion.** The terms distal and proximal are relative to the PMe₃ group. **6, 9**: Ar = 4′-methoxyphenyl; **7, 10**: Ar = 4′-hydroxyphenyl; **8, 11**: Ar = thiophen-2′-yl. Yields: **6-8**: 25–64%; **9–11**: 50–70% dr ~4:1. OTf⁻ = CF₃SO₃⁻. Inset: the stereoselective double-protonation of a deuterobenzene complex followed by EAS of anisole. Tp = (trispyrazolyl)borate.

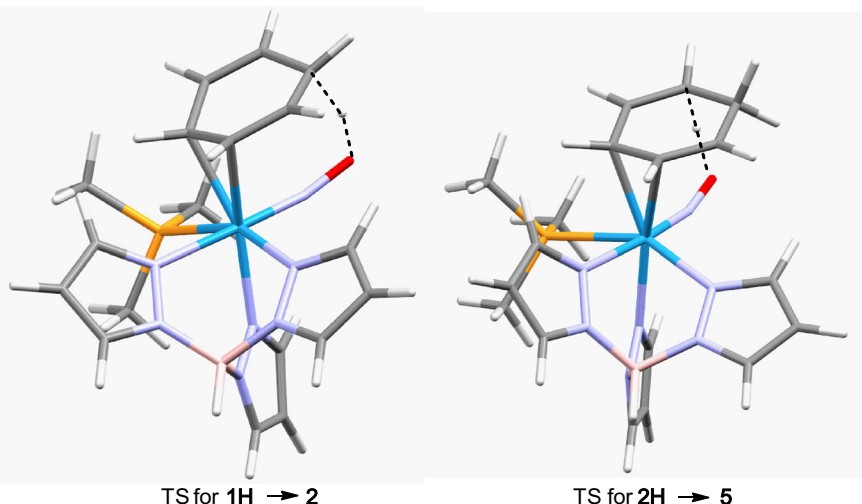

**Fig. 2 | Comparison of first (left) and second (right) intramolecular protonations of benzene. 1H** is the purported NO-protonated conjugate acid of **1**. **2H** is the purported NO-protonated conjugate acid of **2**. Only the first protonation results in stereochemistry supported by experiment.

occurs on the benzene ring to form **5**; NOESY and HSQC NMR data indicate two adjacent diastereotopic methylene groups (Fig. 1). Repeating the reaction in neat DOTf at −78 °C and gradually warming the solution to 0 °C indicates that the initial reaction of **2** and acid generates a paramagnetic complex as indicated by three broad peaks from 5 − 7 ppm. These signals gradually give way to the doubly protonated complex **5 · d₂** as the brown solution turns deep orange. In contrast, if the benzene complex **1** is subjected to HOTf in CD₂Cl₂ at −30 °C, only the monoprotonated complex **2** is formed; this solution evolves to form **5** only after warming to 0 °C. Attempts to isolate the

dicationic complex **5** by precipitation with diethyl ether resulted in decomposition. However, when **5** was generated in situ and treated with anisole, phenol, or thiophene at −30 °C, an electrophilic aromatic substitution (EAS) reaction occurred between the free aromatic and the "carbenium" of **5** proximal to the PMe₃. Addition of the arene occurred anti to the metal to form η²-allyl complexes **6** − **8D**. As demonstrated below, anisole and phenol react at the para carbon (C4′), while thiophene reacts at the α carbon (C2′). This reactivity significantly differs from the precursor η²-benzenium complex **2**, which shows no signs of reactivity with arenes, save for indole[2]. The resulting

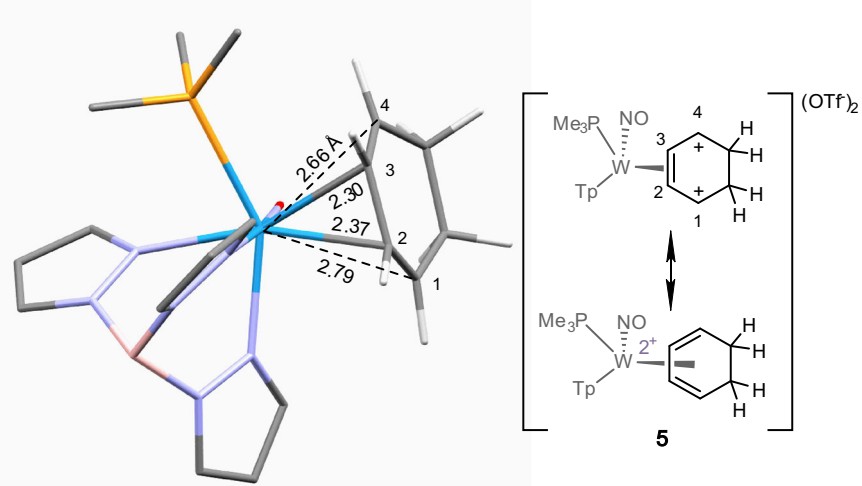

**Fig. 3 | DFT optimized structure of [WTp(NO)(PMe₃)(C₆H₈)]²⁺ (5), the result of the double-protonation of the η²-benzene ligand of 1.** The structure features a highly distorted η⁴-coordinated cyclohexadiene complex.

**Fig. 4 | The tungsten-promoted double-protonation of anisole followed by electrophilic aromatic substitution (EAS), reduction, and addition of cyanide ion.** HOTf = CF₃SO₃H; Tp = trispyrazolylborate. **6,9,15,18**: Ar = 4-methoxyphenyl; **7,10,16,19**: Ar = 4-hydroxyphenyl; **8,11,17,20**: Ar = thiophen-2-yl. Yields for **9–11**: 50–70%.

η²-allyl species **6 - 8D** (also referred to as hyperdistorted η³-allyl[4], or σ-π distorted[5–7]), are heavily weighted toward the conformer with the carbenium carbon distal to the PMe₃[4]. Subsequently, the addition of a second nucleophile (CN⁻)[2,4] resulted in cis-3,6-disubstituted cyclohexene complexes **9 - 11D**. Unfortunately, these products were all

accompanied by roughly 20% of a second isomer, both for the η²-allyl intermediate (**6 - 8 P**) and for the final cyclohexene complex (**9 - 11 P**). These minor products were ultimately characterized (vide infra) as diastereomers of the major cyclohexene products in which the free arene added to the carbenium distal to the PMe₃ and the second

nucleophile added to the proximal allyl carbon. Attempts to improve the diastereoselectivity of this reaction through adjusting temperature, solvent or reaction time were unsuccessful.

When the reaction sequence to generate the anisole addition product **9D** was repeated using the deuterated benzene complex **1**-d₆, the two protons incorporated in the double protonation sequence were identified by two signals appearing for **6D**-d₆ at 3.50 and 1.22 ppm (CD₃CN), corresponding to the 4-endo and 5-exo positions (NOESY). These observations reveal two different mechanisms for protonation: consistent with our earlier studies of monoprotonated benzene, the first protonation occurs syn to the metal (H₄ ₑₙₐₒ)[1], while the second

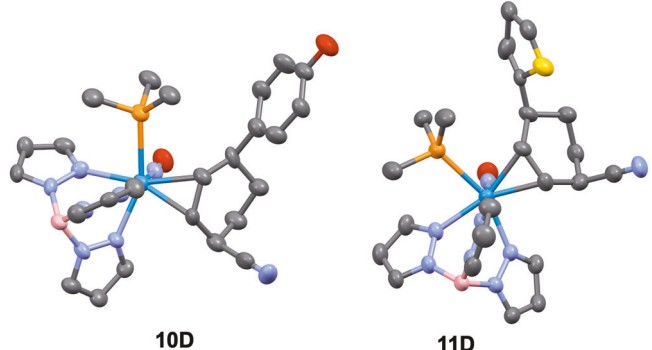

**Fig. 5 | Molecular structure determinations (50% ellipsoids) for the cis-3,6-disubstituted cyclohexene complexes 10D and 11D.** These structures feature the addition of phenol (**10D**) and thiophene (**11D**) to a double-protonated anisole precursor (**14**).

protonation occurs anti (H₅ ₑₓₒ). The ¹H NMR spectrum of **6D**-d₆ indicates that even at high acid concentrations, both protonations are highly regio- and stereoselective, with minimal amounts of proton signal (0-15%) appearing at other positions.

The first protonation of the benzene ring is observed to occur syn to the tungsten[1]. Previously we have demonstrated the ability to methylate the NO ligand of the cyclohexene complex WTp(NO)(PMe₃) (C₆H₁₀)[1], so it stands to reason that the nitrosyl can also be protonated by strong acid, forming **1H**. In turn, DFT calculations demonstrate that a purported transfer of a proton from the NO ligand to the ring (**1H** → **2**; Fig. 2; SI) can occur with a modest transition state energy of 8.2 kcal/ mol and free energy change of −8.8 kcal/mol. It is tempting to ascribe the paramagnetic species (vide supra) observed at −78 °C to the nitrosyl-protonated benzene complex, **1H**. Casting doubt on this hypothesis is the observation that the analogous methylated cyclohexene complex WTp(NOMe)(PMe₃)(C₆H₁₀) is diamagnetic[1]. Calculations further show that an analogous NO-assisted second protonation is possible (TS = 6.1 kcal/mol, ΔG = −13.9 kcal/mol; Fig. 2); however, the double protonation of **1**-d₆ to form **6D**-d₆ unambiguously results in a trans arrangement of the two ring protons, indicating that the second protonation must occur mainly by an intermolecular pathway, anti to the metal. Thus, while we cannot rule out an intermolecular protonation of the η²-benzene for both first and second protonations of **1**, given the high stereoselectivity observed for **6D**-d₆, it seems likely that the first protonation of benzene is intramolecular (NO assisted) while the second is intermolecular.

According to calculations, the double-protonated benzene complex **5** can be considered as a highly distorted η⁴-tungsten(II)-diene complex (Fig. 3), with elongated bond lengths between tungsten and the terminal diene carbons (W-C1 and W-C4: 2.66,

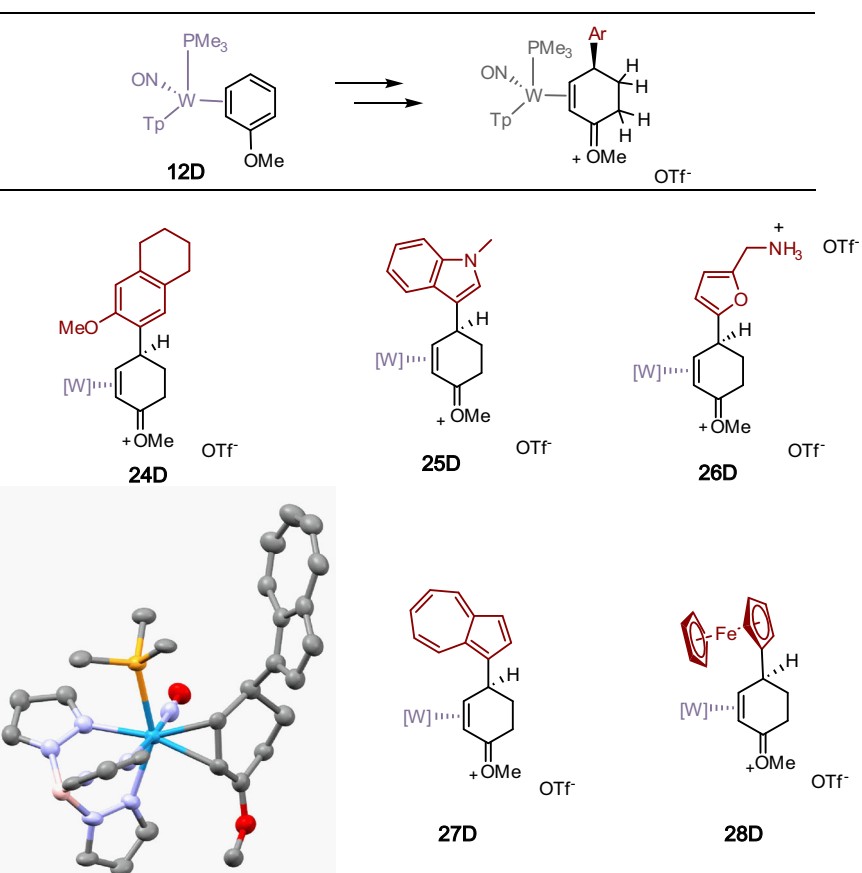

**Fig. 6 | Preliminary scope of aromatic π-nucleophiles that react with the double protonated anisole complex 14D.** Successful additions include benzene derivatives, indole, furans, azulene, and ferrocene. Inset: molecular structure determination of the azulene derivative **27D**.

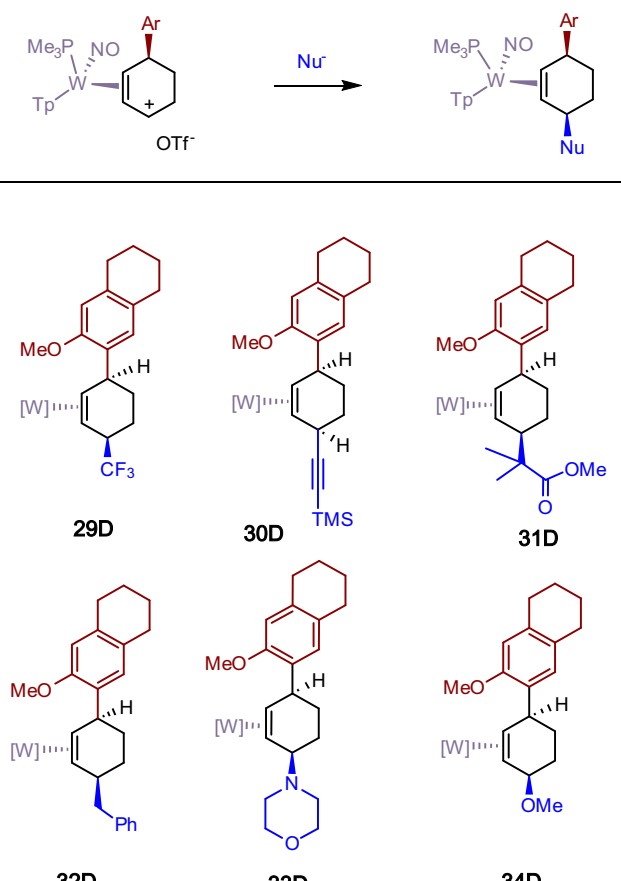

**Fig. 7 | Preliminary examples of cis-3,6-disubstituted cyclohexene complexes prepared from an η²-allyl complex and various nucleophiles.** Nucleophilic reagents include TMSCF₃ (**29D**), trimethylsilylacetylene/BuLi (**30D**), BnMgCl (**32D**), methyl trimethylsilyl dimethylketene acetal (**31D**), morpholine (**33D**), and MeOH/KOH (**34D**).

2.79 Å; cf. W-C2 and W-C3: 2.30, 2.37 Å). These distorted structural features are reminiscent of those seen for the η²-allyl species described earlier. A search of the Cambridge Structural Database[8] failed to identify any analogously distorted η⁴-diene structures. However, the structure of **5** is reminiscent to those found in zirconium and hafnium complexes of η⁴-cyclooctatetrene[9]. The distal carbenium carbon of **5** has the longest bond to the metal (2.79 Å) and might be predicted to be the more reactive site of addition; however, nucleophilic attack occurs predominantly at the proximal carbenium. Such an addition generates η²-allyl species (**6** - **8D**) with the remaining carbenium distal to PMe₃. The distal form (**D**) is known to be several kcal/mol more stable than the isomers resulting from distal addition of the arene (**6**–**8P**)[2]. Hence, we rationalize the kinetic preference for the addition of the arene to the proximal carbenium by invoking a transition state that resembles the product in which the carbenium is distal to the PMe₃.

The observed 4:1 selectivity discouraged us from developing a synthetic method for enantioenriched cyclohexenes using this approach. Hints of analogous reactivity were observed for the molybdenum complex MoTp(NO)(DMAP)(η²-benzene)[10], (**1-Mo**) including spectroscopic evidence for the Mo analog of **6D** (**6D-Mo**). The large-scale preparation of **1-Mo** and spectroscopic data for **6D-Mo** can be found in the SI. However, the high sensitivity of these compounds to acid ultimately discouraged our further investigation.

## Double protonation of anisole

We next considered a modified strategy (Fig. 4) in which an η²-anisole complex would be double-protonated. We reasoned that the methoxy substituent would not only facilitate the double protonation, but also could help direct the aryl addition to the para- carbon of the anisole, analogous to what we have previously observed for anilines[11]. In contrast to our aniline observations, we anticipated that the oxocarbenium could be easily reduced later in the reaction sequence. The tungsten anisole complex **12D** exists in solution as a 3:1 equilibrium with its stereoisomer **12P**[12,13]. However, the 2H-anisolium complex **13D** has been shown to be thermodynamically favored over its proximal analog **13P** ( >20:1), again favoring the oxocarbenium carbon in the distal position[13]. When **13D** was subjected to highly acidic conditions (HOTf/acetonitrile), protonation occurred exclusively at the terminal carbon of the diene fragment (C3) to form the dication **14D** (Fig. 4). We note that changes in the W-P coupling constants are a convenient way to monitor the progress of the reaction: the neutral anisole complex (**12D**) has $J_{WP}$ = 312 Hz, single protonation (**13D**) has $J_{WP}$ = 285 Hz, and double protonation (**14D**) has $J_{WP}$ = 240 Hz. Treating the dication **14D** with phenol, anisole, and thiophene resulted exclusively in the enonium species **15**–**17D**.

Although the anisole complex exists in solution as two interconverting diastereomers (**12D, 12P**), only **12P** is present in the crystalline solid (prepared from solution precipitation)[14]. Adding a cold solution of HOTf in acetonitrile (−30 °C) to a powder of **12** (−30 °C) results exclusively (dr > 20: 1 P/D) in the 2H-anisolium complex **13P**. This compound, like its benzene analogue, can be protonated a second time to form the dicationic complex **14P**, which can then be elaborated into the η²-enonium complexes **15** -**17P** through the additions of anisole, phenol, or thiophene. Subsequent reduction of the η²-enonium complexes (**15**–**17**) for either the P or D series generates the allyl ether complexes **18**–**20**, and this is followed by acid-induced loss of methanol to form the π-allyl complexes **6** - **8**. Finally, treatment with NaCN produces the cis-3,6-substituted cyclohexene complexes **9** - **11**, but in this case, each diastereomer can be generated completely free of the complementary diastereomer (dr > 20: 1; Fig. 5). We note that even though **6** - **8P** favor a conformation in which the carbenium is distal to the PMe₃, steric factors apparently favor addition to the proximal form (Fig. 4). The cyclohexene product is then liberated with the oxidant NOPF₆ to generate the organic compounds 4′-hydroxy-1,2,3,4-tetrahydro-[1,1′-biphenyl]−4-carbonitrile (**21**; 70%), 4′-methoxy-1,2,3,4-tetrahydro-[1,1′-biphenyl]−4-carbonitrile (**22**; 34%), and 4-(thiophen-2-yl)cyclohex-2-ene-1-carbonitrile (**23**; 56%). Although beyond the scope of this preliminary study, we note that the arene complexes **1** and **12** can be prepared in enantioenriched form[2,15,16]. Therefore, while this preliminary report only describes racemic mixtures, either hand of the organic cis-3,6-cyclohexene would be available from an enantioenriched anisole complex.

## Scope of the reaction sequence

In the reaction sequences outlined in Fig. 4, a single regio- and stereoisomer of a cis-3,6-disubstituted cyclohexene complex is obtained (**9** - **11**). In a preliminary screening of arene scope, protonation of **13D** (generated in situ) reacts satisfactorily with other electron-rich heterocycles (indoles and furans) as well as other π systems such as azulene and ferrocene (Fig. 6). Using 6-methoxytetralin as the arene, the resulting oxonium was elaborated into the corresponding π allyl complex analogous to **6**–**8D** (SI; **41D**). This species was then subjected to a range of preliminary screen of carbon nucleophiles including a silylated ester enolate, a silylated acetylide, Rupert's reagent, and benzyl Grignard, as well as the heteronucleophiles morpholine and methoxide. In all cases a cis-3,6-disubstitited cyclohexene complex was produced as a single stereoisomer (**29D**–**34D**; dr > 20:1; Fig. 7). Oxidative decomplexation with NOPF₆ resulted in the free organics **35**–**40** (Fig. 8).

**Fig. 8 | Preliminary examples of cis-3,6-disubstituted cyclohexenes prepared by this method.** These compounds result from the oxidative decomplexation of the corresponding tungsten complexes. dr > 20:1 for all complexes listed.

## Discussion

The Scholl reaction (Friedel-Crafts Arylation) is the coupling of two aromatic rings with the aid of a strong Lewis or Brønsted acid followed by dehydrogenation and is closely related to oxidative aromatic coupling[17]. This historically significant C-C bond forming reaction[17] normally leads to aromatic products, often as oligomeric mixtures, and is driven by the large stabilization gained upon their rearomatization. In contrast, the syntheses of 3,6-disubstituted cyclohexenes such as **21–23** and **35**–**40** is enabled by the tungsten, which inhibits rearomatization of the bound carbocycle. Of note, these compounds have not been reported previously, despite their relatively modest structures. The closest comparisons are 1,4-dihydronapthalene analogs prepared from

a Diels-Alder reaction with benzyne[18], or reaction sequences involving the coupling of aryl halides to cyclohexenes or cyclohexanones. More generally, methods employed to couple aromatics to cycloalkanes typically involve cross-coupling reactions such as Negishi[19], Stille[20], Suzuki[21], and Hiyama couplings[22], but such reactions are more difficult than sp$^2$-sp$^2$ coupling protocols and are often plagued by elimination byproducts. Furthermore, these reactions typically require the use of precious metal catalysts and aryl halides or other suitable aryl precursors. Corey-House[23,24] and Kochi-Schlosser type couplings avoid precious metals but require aryl Grignard reagents[25]. For cases where an organic arene is utilized, strong Lewis acid activators are typically required (Friedel-Crafts)[26]. Lewis and Brønsted acids have successfully

been used in Friedel-Crafts alkylations[26], especially in the case of benzyl electrophiles, where rearrangements of the carbocation intermediate are less of an issue. However, examples carried out with high stereoselectivity are rare due to the fact that the electrophile typically passes through a planar $sp^2$ intermediate[26]. The closest comparisons of EAS reactions related to the current study involve cyclohexadienyliumiron complexes combining with anilines or phenols to generate carbazoles[27,28]. In these studies, the iron complex does not control the stereochemistry of the reaction and cyclohexadienes or arenes are produced. Limited examples of EAS reactions have also appeared in our own work, in the synthesis of γ-substituted enones[29] and tetrahydroindolines[30]. However, in no case previously were we able to couple these reactions to a second nucleophilic addition. A more comprehensive scope of cis-3,6-disubstituted cyclohexenes available by this new method, including enantioenriched variations, will be disclosed in due course.

## Methods

Detailed synthetic procedures and characterizations of all new materials are provided in Supplementary Information document. NMR spectra were obtained on 500, 600, or 800 MHz spectrometers. Chemical shifts are referenced to tetramethylsilane (TMS) utilizing residual $^1H$ or $^{13}C$ signals of the deuterated solvents as internal standards. Phosphorus NMR signals are referenced to 85% $H_3PO_4$ (δ 0.00) using a triphenyl phosphate external standard (δ − 16.58). Chemical shifts are reported in ppm, and coupling constants (J) are reported in hertz (Hz). Infrared (IR) spectra were recorded on a spectrometer as a glaze on a diamond anvil ATR assembly, with peaks reported in $cm^{-1}$. Electrochemical experiments were performed under a nitrogen atmosphere. Cyclic voltammetric data were recorded at ambient temperature at 100 mV/s, unless otherwise noted, with a standard three electrode cell from +1.8 to −1.8 V with a glassy carbon working electrode, tetrabutylammonium hexafluorophosphate (TBAH) electrolyte (~1.0 M), and acetonitrile (MeCN) or tetrahydrofuran (THF) as the solvent. All potentials are reported versus the normal hydrogen electrode (NHE) using cobaltocenium hexafluorophosphate ($E_{1/2}$ = − 0.78, −1.75 V) or ferrocene ($E_{1/2}$ = 0.55 V) as an internal standard. Peak separation of all reversible couples was less than 100 mV. All synthetic reactions were performed in a glovebox under a dry nitrogen atmosphere unless otherwise noted. All solvents were purged with nitrogen prior to use. Deuterated solvents were used as received from Cambridge Isotopes. When possible, pyrazole (Pz) protons of the (trispyrazolyl) borate (Tp) ligand were assigned as "Pz3/5 or Pz4". B-H peaks (around 4 − 5 ppm) in the $^1H$ NMR spectra are not assigned due to their quadrupole broadening; however, confirmation of the BH group is provided by IR data (~2500 $cm^{-1}$). Details of all compound syntheses are provided in the supporting information document.

## Data availability

The authors declare that all data supporting the findings of this study are available within the paper and its supplementary information files. The crystallographic data supporting this work are deposited at the Cambridge Crystallographic Datacenter (CCDC) under the CCDC deposition numbers 2220172-2220177 and 22545778-2254581. These data can be obtained free of charge from The CCDC via www. ccdc.cam.ac.uk/structures. All other data are available from the corresponding author upon request. The data underlying the optimized structures are deposited on figshare (https://doi.org/10.6084/m9.figshare.22734158).

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

## Acknowledgements

Research reported in this publication was supported by the NIGMS of the National Institutes of Health under award number **R01GM132205** (80%) and the University of Virginia (20%). Single crystal X-ray diffraction experiments were performed on a diffractometer at the University of Virginia funded by the NSF-MRI program (CHE-2018870). The content is solely the responsibility of the authors and does not necessarily represent the official views of the National Institutes of Health or the University of Virginia. The authors wish to thank Mary Shingler for the experimental verifications.

## Author contributions

J.T.W.-P., and J.A.S contributed equally to this project. W.D.H., J.A.S., and J.T.W.-P. conceived the project. W.D.H., J.T.W.-P., J.A.S., J.M.B., and J.T.M. designed experiments, prepared samples, and collected NMR and HRMS data. D.A.D. carried out X-ray molecular structure determinations. K.S.W. and M.N.E. carried out DFT calculations. J.A.S., J.T.W.-P., and W.D.H. wrote the manuscript.

## Competing interests

The authors declare no competing interests.
