## [Peer Review File · Nature Communications]

The Double Protonation of Dihapto-Coordinated Benzene Complexes Enables Dearomatization Using Aromatic NucleophilesReviewers' Comments:

Reviewer #1:

Remarks to the Author:

This manuscript by Harman and coworkers details a strategy of using a W complex with a bound arene that gets dearomatized through being doubly protonated. This protonation strategy of a W-benzene complex was the foundation for their 2020 Nature paper. The authors highlight the challenges of selectivity through a direct double protonation strategy. Instead they alter their system to contain a OMe which can direct protonation resulting in >20:1 dr. This is a very clever strategy and has the potential to contribute to the synthesis of a variety of arylated compounds.

The student and postdoctoral researchers should be commended on preparing a variety of challenging compounds as I can tell that a lot of work went into the experimental portion of this work. Despite this, the manuscript almost feels to preliminary, even for a communication. It is unclear to me what the actual impact is on synthesis. This could in part be due to how the authors chose to not display a substrate scope. I can't gather information about what the requirements are for the arene (Ar-H = phenol, anisole, thiophene) nucleophile or the selectivity of it. For example, does phenol add at the ortho, meta, para positions? Is it a mixture? Does Ar-H need to be electron rich? Or can worse nucleophiles work with this highly activated system? Also, why CN as the second nucleophile? Could others be used? I note the last sentence of the manuscript says it will be forthcoming, but I am find myself unable to fully judge the impact of this work without knowing more about the scope of utility. (Note, I am not suggesting that enantioenriched variations are needed for this manuscript, just more understanding about how racemic versions work). I suggest the authors include a discussion of these points in order to allow reviewers to more fully evaluate this manuscript. I do think the manuscript has the potential to be a good fit for Nat. Commun. if these questions can be addressed and the manuscript strengthened but more understanding.

Reviewer #2:

Remarks to the Author:

The contribution by Harman et al. describes a combined experimental (mostly) and computational study of how the double-protonation of a benzene ring coordinated to W can lead to a synthetic strategy to form di-substituted cyclohexene. In relation to previous work by the same authors (refs 1 and 2), the specificity, in terms of novelty, of the present work lies in the reactivity of aromatic compounds as nucleophiles once a double-protonation of the starting complex has been performed. In particular, based on the observation of the presence of two diastereoisomers, the authors do propose a synthetic strategy (Figure 4) that enables to obtain separately either one or the other of the main diastereoisomers. Even though the paper is very clearly written and the chemistry is put in a context where its impact appears as important, it is this referee opinion that the present work does not bring additional conceptual innovation, with respect to previous published works by the authors, to warrant publication as an urgent communication in this journal.

In addition, with a field of expertise in the domain of computational chemistry, I consider that the calculations described in this paper do not bring any specific insight. First the level of calculations (M06/6-31G**, no solvent influence) is well below what is considered nowadays as the basic requirement for calculations. Secondly, the authors did compute transition states for the proton transfer from the NO ligand to the benzene ring. Such calculations yielded a very low barrier for the first, but also for the second protonation. There is thus here a discrepancy with the experimental observations that the authors decided to brush away without even attempting calculations for the inter-molecular protonations. Figure 2 shows dotted lines that are simply assumption of a behavior without any calculations to back-up the claim. Either the authors do the necessary calculations needed to study computationally the competition between inter- and intra-molecular protonation, either they withdraw the calculations from the paper. In their present form they are of no value to understand better the reactivity of the system.

My recommendation is therefore to reject this manuscript.

Reviewer #3:

Remarks to the Author:

This article Smith et al. describes the double protonation of benzene rings coordinated to a zerovalent tungsten complex with strong acids, leading to reactive η^{4} -diene W(II) complex. This achievement is remarkable by itself, but the authors also exploit the reactivity of the latter species to develop a methodology affording *cis*-3,6-disubstituted cyclohexenes by sequential reactions with nucleophiles. The high electrophilicity of the doubly positively charged bis-carbenium species makes it amenable to react with "soft" arene nucleophiles, recalling the postulated ionic mechanism of the Scholl reaction. This contrasts with the reactivity of the singly positively charged η^{2} -allyl W-complex obtained from mono-protonation of coordinated benzene and extensively studied by the same group, showing reactivity only towards harder nucleophiles. In my opinion, the results reported in this manuscript are significant enough to appear in Nat Comm. provided the authors have addressed the following points:

- "the initial reaction of **2** and acid generates a paramagnetic complex as indicated by three broad peaks from 7 - 8.5 ppm". Do the author have an idea of what it could be ? Is that a NO-protonated intermediate they have suggested from DFT ? Can the author try to evidenced this using IR ?
- although difficult to make, the authors do not expand on the synthetic potential of *cis*-3,6-disubstituted cyclohexenes. As for the second nucleophile, only CN⁻ has been considered. Wouldn't it be possible to introduce at the second step another nucleophile that would be suitable to serve as leaving group in subsequent allylic substitution ?
- after reaction of the doubly-charged species **5** with the arene by a Friedel-Crafts type reaction, a Wheland intermediate would be formed carrying a highly acidic proton that is released upon re-aromatization. Can the authors explain where this proton goes ? This allow me to suggest the following experiment: instead of performing the quantitative double protonation of **1**, could the authors let it just react with a slight excess of HOTf (1.1-1.2 equiv) and add one equiv of the arene ? The proton issued by re-aromatization after the Friedel-Crafts reaction should be able to protonate **2** and regenerate **5**.
- please improve the quality of the Figure 2.
- Do the author have tried to record ¹⁸³W NMR ? It would be very interesting to see the impact of protonation on W chemical shifts when going from **1** -> **2** -> **5** and perhaps help confirm the suggested +II oxidation state for W in **5**. As a general comment, since the author heavily rely on NMR to track the reactivity of the W complex, it would be welcome if they can insist a bit more in the manuscript on dramatic changes, signals appearance and disappearance observed in either ¹H or ¹³C NMR.
- "When **13D** was subjected to highly acidic conditions (HOTf/acetonitrile), protonation occurred exclusively at the homoallylic carbon to form the dication **14D**". I am not sure whether "homoallylic" is a correct term here. When looking at the structure of **13D** as depicted in Figure 4, the anisolium moiety does not exhibit strictly speaking allylic positions as it is drawn as a 1,3 diene.
- Please define what is the acid DPhAT
- in their conclusion, could the authors be a little bit more explicit about how their work may impact current knowledge on the Scholl reaction ?

In the supporting information, for the NMR spectra, frequencies, solvent and temperature are missing. The reader has to go to the characterization section to get all the details. A reminder of the structure of the analyzed molecule with atom labels matching assignments would be welcome on each spectrum.

- **5** lacks ³¹P NMR characterization
- spectra concerning the experiment run in neat DOTf are missing

- Figure S-19: can the author explicit what was recorded in the stacked spectra ? Also, the chemical shift axis is missing.

Reviewer #1 (Remarks to the Author):

This manuscript by Harman and coworkers details a strategy of using a W complex with a bound arene that gets dearomatized through being doubly protonated. This protonation strategy of a W-benzene complex was the foundation for their 2020 Nature paper. The authors highlight the challenges of selectivity through a direct double protonation strategy. Instead they alter their system to contain a OMe which can direct protonation resulting in >20:1 dr. This is a very clever strategy and has the potential to contribute to the synthesis of a variety of arylated compounds.

The student and postdoctoral researchers should be commended on preparing a variety of challenging compounds as I can tell that a lot of work went into the experimental portion of this work. Despite this, the manuscript almost feels to preliminary, even for a communication. It is unclear to me what the actual impact is on synthesis. This could in part be due to how the authors chose to not display a substrate scope. I can't gather information about what the requirements are for the arene (Ar-H = phenol, anisole, thiophene) nucleophile or the selectivity of it. For example, does phenol add at the ortho, meta, para positions? Is it a mixture? Does Ar-H need to be electron rich? Or can worse nucleophiles work with this highly activated system? Also, why CN as the second nucleophile? Could others be used? I note the last sentence of the manuscript says it will be forthcoming, but I am find myself unable to fully judge the impact of this work without knowing more about the scope of utility. (Note, I am not suggesting that enantioenriched variations are needed for this manuscript, just more understanding about how racemic versions work). I suggest the authors include a discussion of these points in order to allow reviewers to more fully evaluate this manuscript. I do think the manuscript has the potential to be a good fit for Nat. Commun. if these questions can be addressed and the manuscript strengthened but more understanding.

We have taken the Reviewers comments to heart. The manuscript has been expanded to include a more comprehensive range of "soft nucleophiles" (aromatics) (five additional compounds and crystal structure; Figure 6). We have also added substantially to the range of "hard nucleophiles" (six new compounds; Figure 7), that now include examples of C, N, and O. Examples of final organic products are also included (six new compounds; Figure 8). We note that the scope of reactions is *preliminary* (consistent with Communication format) and we will continue to uncover the full breath of this new synthetic approach.

Per Reviewer 1's request, we have also clarified in the text the position of the electrophilic substitution reaction for anisole, phenol, and thiophene.

Reviewer #2 (Remarks to the Author):

The contribution by Harman et al. describes a combined experimental (mostly) and computational study of how the double-protonation of a benzene ring coordinated to W can lead to a synthetic strategy to form di-substituted cyclohexene. In relation to previous work by the same authors (refs 1 and 2), the specificity, in terms of novelty, of the present work lies in the reactivity of aromatic compounds as nucleophiles once a double-protonation of the starting complex has been performed. In particular, based on the observation of the presence of two diastereoisomers, the authors do propose a synthetic strategy (Figure 4) that enables to obtain separately either one or the other of the main diastereoisomers. Even though the paper is very clearly written and the chemistry is put in a context where its impact appears as important, it is this referee opinion that the present work does not bring additional conceptual innovation, with respect to previous published works by the authors, to warrant publication as an urgent communication in this journal.

We respectfully disagree with Reviewer 2. As the other two reviewers recognized, the strategy of a double-protonation of benzene is entirely new, and the impact is potentially far-reaching. Contrary to what Reviewer 2 writes, double-protonation was never mentioned in our preliminary work with benzene (refs 1 and 2).

*In addition, with a field of expertise in the domain of computational chemistry, I consider that the calculations described in this paper do not bring any specific insight. First the level of calculations (M06/6-31G**, no solvent influence) is well below what is considered nowadays as the basic requirement for calculations. Secondly, the authors did compute transition states for the proton transfer from the NO ligand to the benzene ring. Such calculations yielded a very low barrier for the first, but also for the second protonation. There is thus here a discrepancy with the*

experimental observations that the authors decided to brush away without even attempting calculations for the inter-molecular protonations....

In a recent paper, (*J. Am. Chem. Soc.* 2020, 142, 38, 16437) we tested a variety of functionals and basis sets, and the ones used in this study performed the best in matching crystal structure data. It is a simple matter to include a solvent correction, which we have now done (the numbers are virtually the same before and after solvent correction), but what Reviewer 2 is asking for is to model the intermolecular proton transfer, which is virtually impossible without detailed knowledge of the acid. The acid (triflic acid) is almost certainly a cluster involving solvent, and depending on how we model it, the change in free energy and transition state will vary considerably. Such detailed analysis is well beyond the scope of this study. Nonetheless, we feel that modeling the ground state of the double-protonated benzene is informative, as is establishing that the intramolecular NO assisted proton transfer is *possible*. Given the exclusive endo-selective nature of the first protonation, NO-assisted proton transfer is probable mechanism (but of course, we cannot prove it).

....*Figure 2 shows dotted lines that are simply assumption of a behavior without any calculations to back-up the claim. Either the authors do the necessary calculations needed to study computationally the competition between inter- and intra-molecular protonation, either they withdraw the calculations from the paper. In their present form they are of no value to understand better the reactivity of the system.*

We agree that a detailed discussion of inter- versus intramolecular proton transfer is premature. We have simplified Figure 2, removing the reaction coordinate diagrams and keeping only the structures of the transition states for NO proton transfer (these low barriers are remarkable, and not at all intuitive to organometallic chemists).

Reviewer #3 (Remarks to the Author):

This article Smith et al. describes the double protonation of benzene rings coordinated to a zerovalent tungsten complex with strong acids, leading to reactive η^4 -diene W(II) complex. This achievement is remarkable by itself, but the authors also exploit the reactivity of the latter species to develop a methodology affording cis-3,6-disubstituted cyclohexenes by sequential reactions with nucleophiles. The high electrophilicity of the doubly positively charged bis-carbenium species makes it amenable to react with "soft" arene nucleophiles, recalling the postulated ionic mechanism of the Scholl reaction. This contrasts with the reactivity of the singly positively charged η^2 -allyl W-complex obtained from mono-protonation of coordinated benzene and extensively studied by the same group, showing reactivity only towards harder nucleophiles. In my opinion, the results reported in this manuscript are significant enough to appear in Nat Comm. provided the authors have addressed the following points:

• *"the initial reaction of 2 and acid generates a paramagnetic complex as indicated by three broad peaks from 7 - 8.5 ppm". Do the author have an idea of what it could be ? Is that a NO-protonated intermediate they have suggested from DFT ? Can the author try to evidenced this using IR ?*

The postulated protonated NO complex is almost certainly diamagnetic, analogous to the methylated NO derivative (which we have characterized by NMR). We address this in the revised manuscript. It's possible that the paramagnetic intermediate is a precursor to the double protonation, but the reaction conditions under which this species is formed (concentrated triflic acid at -78 C) make it very difficult to collect additional spectroscopic information with any confidence that we are catching this particular intermediate. Whatever the nature of the paramagnetic material, it is transient and does not detract from the central thesis of the paper (double protonation of an arene). We included this observation as it may be relevant to some future study, but can remove it if the uncertainty of its identity detracts from the manuscript.

• *although difficult to make, the authors do not expand on the synthetic potential of cis-3,6-disubstituted cyclohexenes. As for the second nucleophile, only CN^- has been considered. Wouldn't it be possible to introduce at the second step another nucleophile that would be suitable to serve as leaving group in subsequent allylic substitution ?*

This is a good idea! But even better, we can add a full range of nucleophiles directly to the allylic intermediates. We have included a new figure (Figure 7) showing this preliminary scope.

• after reaction of the doubly-charged species **5** with the arene by a Friedel-Crafts type reaction, a Wheland intermediate would be formed carrying a highly acidic proton that is released upon re-aromatization. Can the authors explain where this proton goes? This allows me to suggest the following experiment: instead of performing the quantitative double protonation of **1**, could the authors let it just react with a slight excess of HOTf (1.1-1.2 equiv) and add one equiv of the arene? The proton issued by re-aromatization after the Friedel-Crafts reaction should be able to protonate **2** and regenerate **5**.

An excellent idea. When we tried 1.2 eq of acid, however, we were unable to achieve the arene reaction. It's possible that the NO of the product complex is the most basic site in the system, and it effectively prevents the propagation that Reviewer 3 envisioned.

• please improve the quality of the Figure 2.

Figure 2 has been reworked

• Do the authors have tried to record ^{183}W NMR? It would be very interesting to see the impact of protonation on W chemical shifts when going from **1** → **2** → **5** and perhaps help confirm the suggested +II oxidation state for W in **5**. As a general comment, since the authors heavily rely on NMR to track the reactivity of the W complex, it would be welcome if they can insist a bit more in the manuscript on dramatic changes, signals appearance and disappearance observed in either ^1H or ^{13}C NMR.

Unfortunately, we do not currently have the capabilities of recording ^{183}W NMR. However, ^{31}P - ^{183}W coupling (via ^{31}P NMR) is an exceptionally good way to follow these reactions and we have now included a sentence explaining what features to look for.

• "When **13D** was subjected to highly acidic conditions (HOTf/acetonitrile), protonation occurred exclusively at the homoallylic carbon to form the dication **14D**". I am not sure whether "homoallylic" is a correct term here. When looking at the structure of **13D** as depicted in Figure 4, the anisole moiety does not exhibit strictly speaking allylic positions as it is drawn as a 1,3 diene.

This wording has been corrected.

• Please define what is the acid DPhAT

DPhAT is now defined in text.

• in their conclusion, could the authors be a little bit more explicit about how their work may impact current knowledge on the Scholl reaction?

The discussion section has now been expanded to include this point.

In the supporting information, for the NMR spectra, frequencies, solvent and temperature are missing. The reader has to go to the characterization section to get all the details. A reminder of the structure of the analyzed molecule with atom labels matching assignments would be welcome on each spectrum.

- **5** lacks ^{31}P NMR characterization
- spectra concerning the experiment run in neat DOTf are missing
- Figure S-19: can the authors be explicit what was recorded in the stacked spectra? Also, the chemical shift axis is missing.

The following changes have been made to the SI:

- Added spectra and experimentals for compounds **24D-41D**
- Included NMR spectrometer frequencies, solvents, and temperatures to each spectrum
- Include atom labels for **14P** and **14D**
- Remade Figure S-19 (page 22) to show the chemical shift axis and what each spectrum corresponds to
- Added Figure S-42 (page 66) which shows a ^1H NMR of the initial treatment of **1** with neat DOTf and what's observed after warming to 0 °C

- Added updated crystal data.
- Added revised DFT calculations section including solvent corrected values

Reviewers' Comments:

Reviewer #2:

Remarks to the Author:

As far as the decision to publish this work in Nature Communications, obviously my opinion and that of the authors do differ but my field of expertise being Computational Studies, I do not feel I am in a position to have a strong opinion. I leave thus the final decision to the editor.

Concerning what is brought in this work by the computational study, the revised version does a better job at clearly presenting the actual facts and does acknowledge that a final conclusion about the protonation mechanism can not be inferred from the calculations only. Hence this referee is legitimate to ask the simple question : what are the calculations useful for in the present context ? To my understanding they do not bring any significant insight apart the possibility to get geometrical information on fleeting intermediates (Figure 3). As the paper is very clearly written and as the chemistry is very nice and has been appreciated by the other referees, I would not want my opinion to be the reason to reject this manuscript. However I wanted to be very clear on my appreciation on the computational study, even though it has been properly carried out.

Reviewer #3:

Remarks to the Author:

My comments and those of the other reviewers have been taken into account in this revised version. I congratulate the authors for seriously addressing the issues we raised, especially concerning the scope/synthetic potential.

The article can be published in its present form without any further modification.

Point-by-point response to the reviewer's comments. NCOMMS-23-00968A

REVIEWERS' COMMENTS

Reviewer #2 (Remarks to the Author):

As far as the decision to publish this work in Nature Communications, obviously my opinion and that of the authors do differ but my field of expertise being Computational Studies, I do not feel I am in a position to have a strong opinion. I leave thus the final decision to the editor.

Concerning what is brought in this work by the computational study, the revised version does a better job at clearly presenting the actual facts and does acknowledge that a final conclusion about the protonation mechanism can not be inferred from the calculations only. Hence this referee is legitimate to ask the simple question : **what are the calculations useful for in the present context ?** To my understanding they do not bring any significant insight apart the possibility to get geometrical information on fleeting intermediates (Figure 3). As the paper is very clearly written and as the chemistry is very nice and has been appreciated by the other referees, I would not want my opinion to be the reason to reject this manuscript. **However I wanted to be very clear on my appreciation on the computational study, even though it has been properly carried out.**

Response: This reviewer seems to be at peace with the current manuscript. Again, our main reason for including the computational structures and transition states is to show that the hypothesized intramolecular proton transfer from the nitrosyl ligand is feasible and that it explains the observed stereochemistry. No further changes have been made to the revised text.

Reviewer #3 (Remarks to the Author):

My comments and those of the other reviewers have been taken into account in this revised version. I congratulate the authors for seriously addressing the issues we raised, especially concerning the scope/synthetic potential. **The article can be published in its present form without any further modification.**

Response: This reviewer seems to be pleased with the current manuscript. No further changes have been made to the revised text.